# Loss of CD34 Expression within an Interstitial Dermal Lymphoid Cell Infiltrate Is a Helpful Clue to the Diagnosis of Morphea

**Maged Daruish** [1] , **Anoud Zidan** [1], **Danielle T. Greenblatt** [2] **and Catherine M. Stefanato** [1,*]

1 Department of Dermatopathology, St John's Institute of Dermatology, Guy's and St Thomas' NHS Foundation Trust, London SE1 7EH, UK

2 Pediatric Dermatology, St John's Institute of Dermatology, Guy's and St Thomas' NHS Foundation Trust, London SE1 9RT, UK

* Correspondence: catherinestefanato@gmail.com

**Abstract:** A dermal interstitial lymphocytic infiltrate may represent a diagnostic challenge, particularly if the clinical history is not provided. We present three cases within the histological spectrum of morphea in which the immunohistochemical marker CD34 was helpful in confirming the diagnosis.

**Keywords:** morphea; dermal dendritic cells; CD34 immunohistochemistry

## 1. Introduction

Morphea is an autoimmune disorder that causes fibrosis of the skin and is most frequently seen in young girls. The pathogenesis is not fully elucidated; however, it appears that the fibroblasts are activated by various environmental stimuli in the presence of genetic predisposition [1].

The most common histological features are collagen homogenization, sclerosis and thickening of the dermis, and loss of peri-adnexal fat or reduced appendages of the skin [1]. Other practical helpful clues include eccrine glands located higher up in the dermis, square-shaped biopsies with a 90-degree angle at each corner of the biopsy, and the line-sign, where the interface between the sclerosed dermis and subcutis appears linear at scanning magnification [2].

However, the diagnosis of morphea histologically can be challenging, particularly in the early inflammatory stage when it might lack the characteristic stromal changes and manifest with a non-specific perivascular and interstitial lymphocytic cell infiltrate. Accordingly, about half of the biopsies may be non-diagnostic [1,3]. In addition, distinguishing inflammatory morphea from the late inactive stage is difficult both clinically and histologically [1].

A negative CD34 staining in the reticular dermis has been previously demonstrated in biopsies from morphea lesions [4,5]. We present three cases of morphea at different histological stages, in which CD34 immunohistochemistry was helpful in supporting the diagnosis along with clinicopathological correlation.

## 2. Case Report

### 2.1. Case (1)

A nine-year-old girl presented with a ten-month history of a linear erythematous rash with mild pruritus on both her forearms (Figure 1A). The child was otherwise healthy. The autoimmunity panel, including ANA and ENA, was negative. A punch biopsy was obtained from her forearm. Histopathological evaluation revealed a mild to moderately dense superficial and deep lymphocytic infiltrate with scattered plasma cells. There was some loss of the peri-eccrine fat and a subtle thickening of the collagen bundles (Figure 1B,C).

CD34 immunohistochemistry showed reduction of expression in a geographical pattern in the reticular dermis (Figure 1D). The diagnosis of linear morphea in the early inflammatory stage was made by clinicopathological correlation.

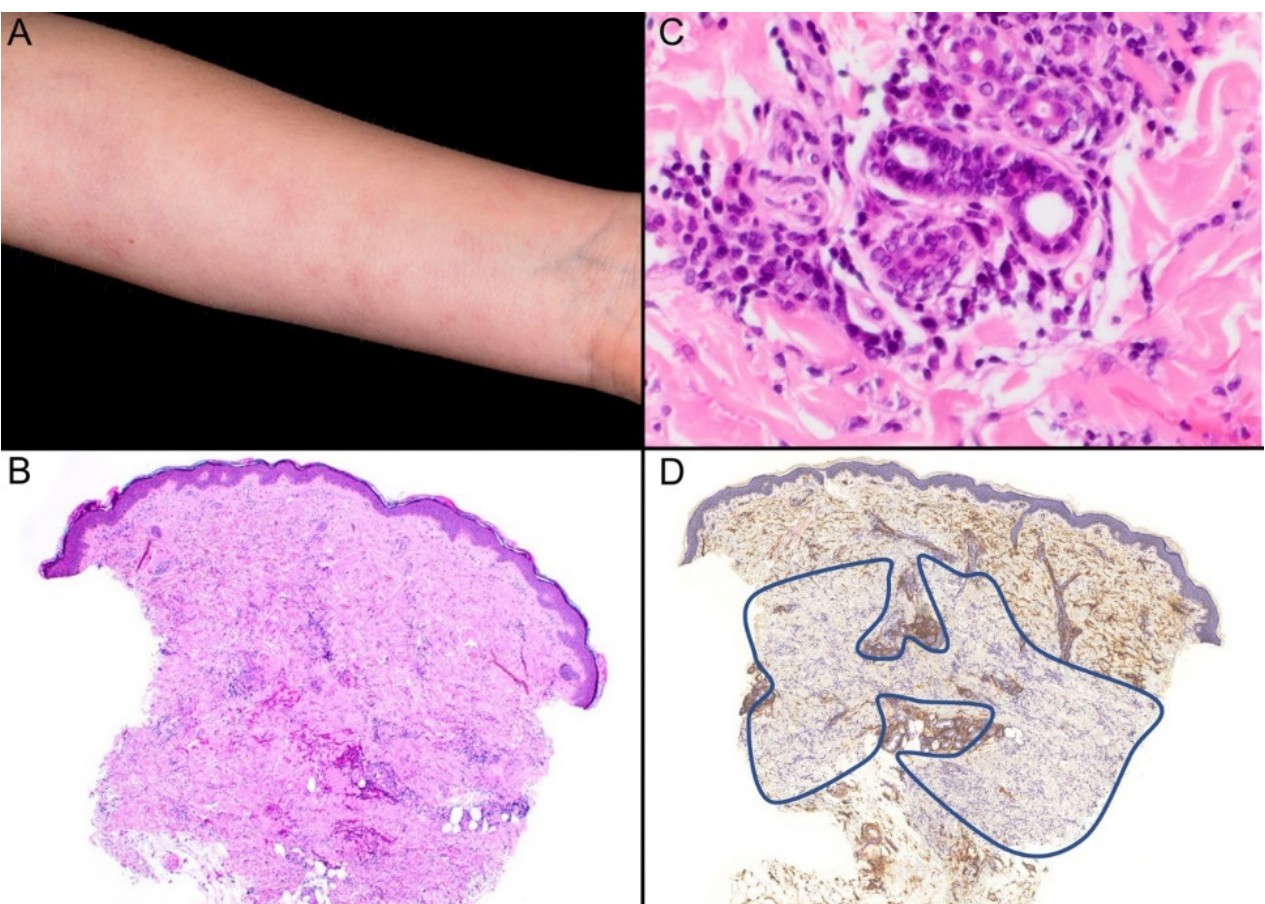

**Figure 1.** (**A**) Subtle erythematous rash on the forearm in a linear pattern. (**B**) Histopathologic examination of the skin punch biopsy reveals a sparse interstitial inflammatory cell infiltrate. Hematoxylin and eosin, (H&E) ×40. (**C**) Loss of the fat around eccrine glands, mild thickening of the collagen bundles, and a mild lymphocytic inflammatory cell infiltrate with plasma cells (H&E) ×200. (**D**) CD34 expression is reduced in the lower reticular dermis in a geographical pattern (as demarcated by the blue line) ×40.

*2.2. Case (2)*

A punch biopsy from a 57-year-old female was received for evaluation. The biopsy was sent from general practice with the minimal clinical detail of a rash on the left hip. Microscopic evaluation revealed hyperkeratosis with follicular plugging and acanthosis of the epidermis. There was homogenization of the collagen in the superficial dermis, while the mid-to-deep dermis showed a mild interstitial lymphoid cell infiltrate with some thickening of the collagen bundles (Figure 2A–C). Loss of CD34 expression was seen throughout the dermis (Figure 2D). The histological features were found to be consistent with lichen sclerosus/morphea overlap.

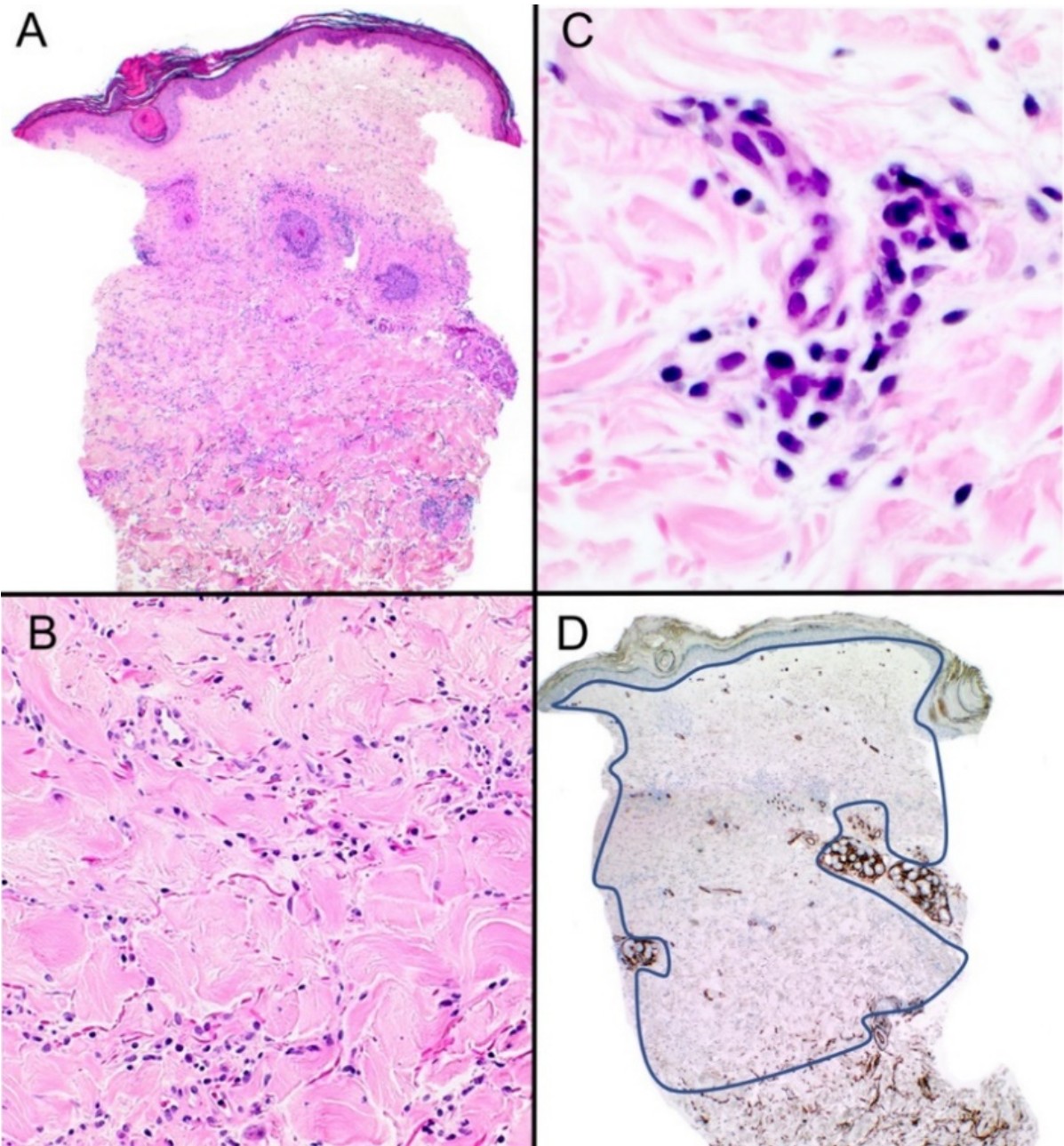

**Figure 2.** (**A**) Scanning magnification of this punch biopsy with hyperkeratosis, follicular plugging, and hyalinization of collagen in the superficial dermis (H&E) ×40. (**B**) Higher magnification shows an interstitial inflammatory cell infiltrate composed predominantly of lymphocytes (H&E) ×100. (**C**) Plasma cell also seen in the infiltrate (H&E) ×200 (**D**) There is diffuse loss of CD34 expression throughout the dermis (as demarcated by the blue line) ×40.

*2.3. Case (3)*

A 58-year-old female presented with a one-year history of a patch of hyperpigmentation on the left upper inner arm (Figure 3A). The patient complained of mild itching and soreness in the area. She did not have any relevant past medical history. Two punch biopsies were taken with the clinical suspicion of lichen planus pigmentosus. Both biopsies showed similar features, including thickening and hyalinization of the collagen bundles, loss of adnexal structures, and a mild perivascular and interstitial infiltrate of lymphocytes with a few plasma cells (Figure 3B,C). CD34 showed loss in both the papillary and reticular dermis (Figure 3D). The diagnosis of morphea in its late stage was made.

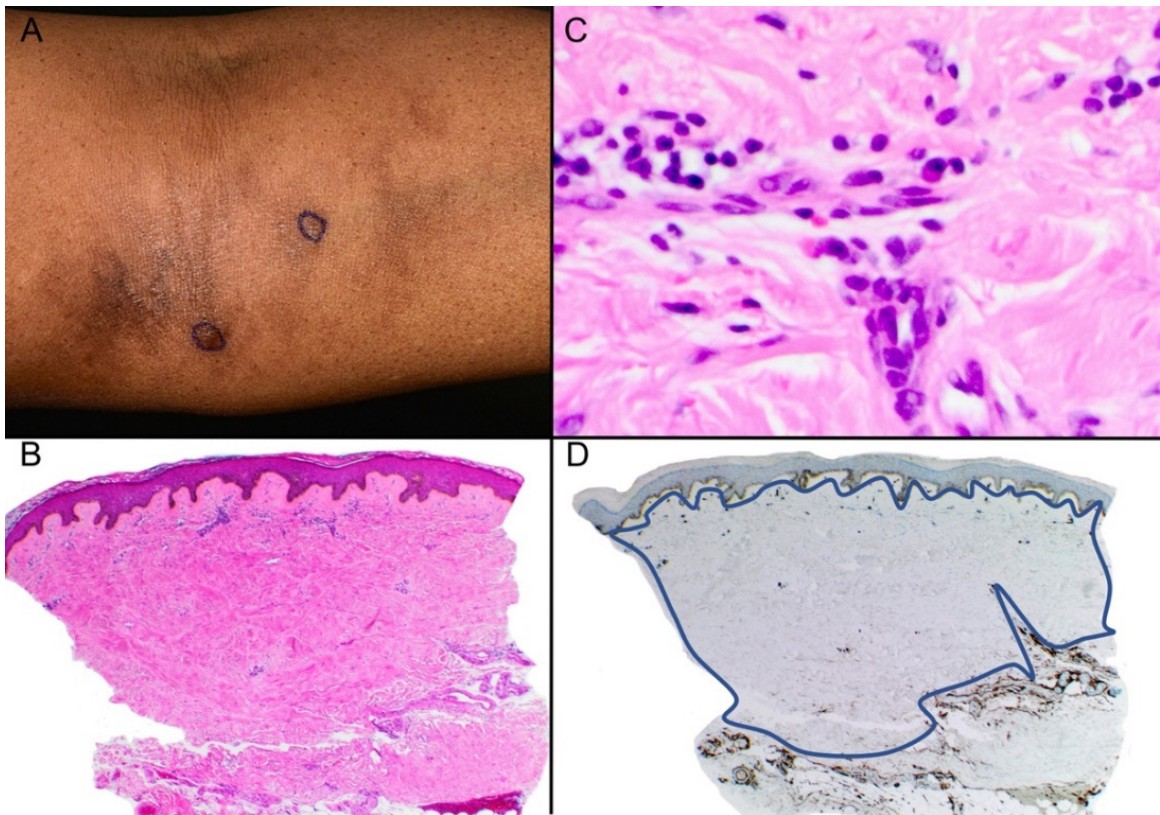

**Figure 3.** (**A**) Hyperpigmented area seen in the left upper inner arm. (**B**) Histopathologic examination shows hyperkeratosis and acanthosis with diffuse thickening of the superficial and deep dermis with loss of adnexal structures. (H&E) ×40. (**C**) Thickening of the collagen bundles with hyalinization is present with a mild interstitial cell infiltrate composed of lymphocytes and plasma cells (H&E) ×400. (**D**) CD34 expression is diffusely absent in the dermis (as demarcated by the blue line) ×40.

## 3. Discussion

CD34+ Dermal dendritic cells (DDCs) are normally distributed in the reticular dermis and around the adnexa. They are postulated to have functions in wound healing and the maintenance of dermal architecture [4].

Multiple studies have noted that there is a reduction of dermal dendritic cells in morphea. Skobieranda et al. [5] demonstrated a reduction in CD34 expression in 26 morphea biopsies compared to 11 specimens from normal skin. This was associated with an increase in vimentin and Factor XIIIa expression. The authors incidentally found similar results in scar tissue. Thus, they hypothesized that CD34 DDCs may play a role in the regulation of fibroblast activity.

A similar study, conducted on a larger population (50 patients and 50 controls), was concluded with similar results. In addition, they found elevated SMA expression, which they explained as a phenotypic shift from CD34 DDCs to myofibroblasts [4].

Reduction or loss of CD34 expression has also been described in similar sclerosing disorders such as lichen sclerosus, systemic sclerosis, lipodermatosclerosis, and sclerodermoid graft-versus-host disease, as well as scars [5–8]. Accordingly, it should not be used as a sole diagnostic criterion and should be combined with clinical data in this setting.

The differential diagnosis of lymphocyte-predominant interstitial inflammatory cell infiltrate is broad and varies from Schamberg's disease to interstitial mycosis fungoides (IMF) [3]. Indeed, distinguishing early morphea from these disorders can be quite difficult [3].

In this setting, loss of CD34 expression and plasma cells in the infiltrate point towards the diagnosis of morphea [3]. In a recent study, plasma cells were always found in morphea specimens even when collagen homogenization was absent [1].

In Schamberg's disease, extravasation of erythrocytes and hemosiderin deposition are helpful findings [3]. Early erythema migrans may have similar features to morphea, with a perivascular and interstitial superficial and deep lymphoplasmacytic cell infiltrate; however, frequent extension of the infiltrate in the subcutaneous fat is typically present. Neutrophils and eosinophils may also be seen [3]. As for IMF, the presence of lymphocytic atypia is the main differentiating feature, and the lymphocytes are usually of the cytotoxic CD8+ phenotype [9,10].

It is important to emphasize that the loss of CD34 staining in the dermis is not entirely diagnostic and should not be used in differentiating morphea from scars and other cutaneous diseases in which sclerosis is the main pathology.

However, in the appropriate clinical setting, we underscore that this can be a very valuable clue for differentiating morphea from other lymphocytic-predominant interstitial infiltrates, especially in early and subtle clinical cases such as our first case or in cases with minimal clinical information as in our second case. In the former, the absence of CD34 positivity was mainly in the lower reticular dermis, while in the second and third cases, the reticular dermis was diffusely negative for CD34. This may be explained by the previous observation that the degree of CD34 expression is inversely correlated to the degree of dermal sclerosis in morphea [4].

Future studies are necessary to determine the sensitivity of CD34 loss in morphea cases and whether there is a pattern that can be correlated with the stage of the disease.

**Author Contributions:** Writing—original draft preparation, M.D.; writing—review and editing, M.D., C.M.S. supervision, A.Z., D.T.G., C.M.S. All authors have read and agreed to the published version of the manuscript.

**Funding:** This research received no external funding.

**Institutional Review Board Statement:** Not applicable.

**Informed Consent Statement:** Written informed consent has been obtained to publish this paper.

**Conflicts of Interest:** The authors declare no conflict of interest.

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
