# Peer review of "Loss of CD34 Expression within an Interstitial Dermal Lymphoid Cell Infiltrate Is a Helpful Clue to the Diagnosis of Morphea"

_dermatopathology, doi:10.3390/dermatopathology10010010_

Round 1

Reviewer 1 Report

CD34 should be considered one more clue in the diagnsis of early morphea togheter with the presence of plasmacells, as shown in the pictures of present case, and with the other well known features , combined with clinical data.

1) Main question addressed is the usefulness of CD34 in early morphea diagnosis.
2) The topic is original
3) The introduction of one more clue for the diagnosis of morphea according with clinical histological correlations
5) The conclusions consistent with the evidence and arguments presented and they address the main question posed
6) Some more references are requested
7) The figures are very clear and significative and in my opinion should be underlined the presence of plasmacells in a better way.

Author Response

We would like to thank the reviewers for taking the time to review this paper and
appreciate their valuable comments which we have now incorporated into the manuscript.
We appreciate the opportunity to respond to the reviewers’ comments as outlined
below.
Reviewer 1:
- Some more references are requested
REPLY: More references have been added
- The figures are very clear and significative and in my opinion should be underlined
the presence of plasmacells in a better way:
REPLY: Higher magnification pictures have been provided for each histopathology figure
highlighting the presence of plasma cells
Reviewer 2:
- Clinicopathological features are great, but first patient, figure 1a lesions are barely
visible. If possible, select a clearer photograph to illustrate the case.
REPLY: In accordance, we have replaced figure 1a with a better clinical photograph
- Results on the changes of CD34 in morphea, should be clearer expressed
REPLY: In response we have now expanded the manuscript to further discuss the patterns
we noticed in our cases in page 5 lines: 121-125
- Other clues for diagnosis of morphea as the square punch or the linear rope should
be added
REPLY: These signs have been incorporated into the manuscript and are now discussed in
page 5 lines: 105-106
- Differential diagnosis of lymphocyte-predominant interstitial inflammatory cells can
be improved by clinicopathological correlation as well as by immunohistochemistry.
This should be cleared up.
REPLY: We have discussed the differential diagnosis further in page 5 lines: 103-114
- Finally, state author´s opinion on how it can be more useful to ask for CD34 in this
type of cases, to increase the didactic value of the cases presented

REPLY: We have added the last two paragraphs in which we further discuss the scenarios in
which CD34 immunohistochemistry would be a valuable tool in the histopathological
diagnosis of morphea.
We feel that, thanks to your constructive comments the manuscript has now substantially
improved, and hope it will be acceptable for publication.
In thanking you, we look forward to hearing from you,
Catherine M. Stefanato, MD, FRCPath
Consultant in Dermatopathology
St John's Department of Dermatopathology
Corresponding Author
Maged Dariush, MSc
Senior Clinical Fellow in Dermatopathology

Reviewer 2 Report

Clinicopathological features are great, but first patient, figure 1a lesions are barely visible. If possible, select a clearer photograph to illustrate the case.

Results on the changes of CD34 in morphea, should be clearer expressed.

Other clues for diagnosis of morphea as the square punch or the linear rope should be added.

Differential diagnosis of lymphocyte-predominant interstitial inflammatory cells can be improved by clinicopathological correlation as well as by immunohistochemistry. This should be cleared up. 

Finally, state author´s opinion on how it can be more useful to ask for CD34 in this type of cases, to incrase the didactic value of the cases presented-

Author Response

(The authors gave the same response as above.)
